# Subjective risk and associated electrodermal activity of a self-driving car passenger in an urban shared space

**Jeffery Petit[1], Camilo Charron[1,2], Franck Mars[1] ***

**1** École Centrale Nantes, CNRS, LS2N, UMR 6004, Nantes Université, Nantes, France, **2** Université de Rennes 2, Rennes, France

\* franck.mars@ls2n.fr

**Data Availability Statement:** The dataset and the algorithm used in this study are available as an R package that can be downloaded from https://gitlab.univ-nantes.fr/mars-f/bnscore.

## Abstract

Shared spaces are urban areas without physical separation between motorised and non-motorised users. Previous research has suggested that it is difficult for users to appropriate these spaces and that the advent of self-driving cars could further complicate interactions. It is therefore important to study the perception of these spaces from the users' perspectives to determine which conditions may promote their acceptance of the vehicles. This study investigates the perceived collision risk of a self-driving car's passenger when pedestrians cross the vehicle's path. The experiment was conducted with a driving simulator. Seven factors were manipulated to vary the dynamics of the crossing situations in order to analyse their influence on the passenger's perception of collision risk. Two measures of perceived risk were obtained. A continuous subjective assessment, reflecting an explicit risk evaluation, was reported in real time by participants. On the other hand, their skin conductance responses, which reflects implicit information processing, were recorded. The relationship between the factors and the risk perception indicators was studied using Bayesian networks. The best Bayesian networks demonstrate that subjective collision risk assessments are primarily influenced by the factors that determine the relative positions of the vehicle and the pedestrian as well as the distance between them when they are in close proximity. The analysis further reveals that variations in skin conductance response indicators are more likely to be explained by variations in subjective assessments than by variations in the manipulated factors. These findings could benefit the development of self-driving navigation among pedestrians by improving understanding of the factors that influence passengers' feelings.

## 1. Shared space and vehicle–pedestrian interaction

The emergence of shared spaces introduces complexity to the navigation of autonomous vehicles. As described by Hamilton-Baillie [1], shared spaces are areas where all forms of segregation between pedestrians and vehicles (e.g., road markings, traffic signals, signs, kerbs and barriers) are minimised or eliminated. Shared spaces are often located in high-traffic areas to improve traffic flow and interaction between pedestrians and vehicles. Such spaces are more

**Funding:** This study was supported by the French National Research Agency (Agence Nationale de la Recherche, HIANIC project, Grant no. ANR-17-CE22-0010; awarded to F.M.). The funders had no role in study design, data collection and analysis, decision to publish, or preparation of the manuscript.

**Competing interests:** The authors have declared that no competing interests exist.

conducive to pedestrian activity, and the absence of signals forces pedestrians and drivers to be more aware of the movement dynamics of surrounding users. In the absence of explicit rules, the fluidity of traffic flow generally relies on cultural norms and informal social protocols.

According to the UK Department for Transport [2], shared spaces are designed to improve pedestrian travel and comfort by reducing the dominance of motorised vehicles. The introduction of such spaces is challenging for both pedestrians and vehicles. According to Hamilton-Baillie [1], shared spaces imply an increase in collision risks, with the corollary that drivers and pedestrians are encouraged to monitor their surroundings more actively. However, Hamilton-Baillie [1] recognises that this requires behavioural adaptations by all agents. Each party must progressively learn to use the available space. Kaparias et al. [3] find that drivers feel less inclined to travel in dense pedestrian areas, particularly in the presence of children or elderly people. Moody & Melia [4] analyse traffic in a shared space in Ashford (UK), revealing that pedestrians tend to yield to vehicles and return to their pre-shared space travel zones.

## 1.1. Autonomous navigation among pedestrians

The challenge of shared spaces increases with autonomous vehicles. Studies have indicated that pedestrians express some reservations about interacting with an autonomous vehicle. For example, Jayaraman et al.'s [5] experiment in an immersive virtual environment demonstrated that pedestrians are more reticent to cross the path of an autonomous vehicle in the absence of a crosswalk. In addition, it has been shown that pedestrians are less likely to cross a street in front of an autonomous vehicle, when they recognize it as one [6, 7]. The vulnerability experienced by pedestrians requires a significant change in the behavioural programming of vehicles that must share their circulation space. Shared spaces therefore pose a challenge for both pedestrians and self-driving car passengers and programmers must develop the interaction between them to ensure a smooth traffic flow that preserves the safety of all concerned.

In the absence of a social relationship between the driver and the surrounding pedestrians (e.g., eye contact or simple gaze orientation), researchers are interested in discovering how a self-driving car can warn pedestrians of its intentions. Equipping an autonomous vehicle with an external human–machine interface (eHMI) is beneficial to its interactions with pedestrians. For example, Faas et al. [8] tested different eHMIs and revealed that providing information about the vehicle's next movements can improve the interaction between pedestrians and autonomous vehicles. Métayer and Coeugnet [9] also demonstrated that pedestrians are more likely to cross in front of an autonomous vehicle when the latter is equipped with an eHMI.

Kyriakidis et al. [10] interviewed 12 expert researchers in the field of human factors in automated driving, revealing that in-vehicle communication is another aspect that must be developed. This type of communication, referred to as *internal human–machine interface* (iHMI), is essential for the passengers of autonomous vehicles to feel safe [11]. Nevertheless, even with adequate in-vehicle communication, the autonomous vehicle's adapted driving behavioural programming remains one of the main factors that influence passengers' experiences. This key point conditions the acceptance of autonomous vehicles by pedestrians outside the vehicle and passengers inside.

## 1.2. Passenger feelings

Studies have shown that speed, acceleration and safe distances from other road users significantly affect passengers' feelings [12–14]. When the autonomous vehicle must navigate among pedestrians, its speed and trajectory are crucial factors that determine the vehicle's safety margins with pedestrians. Gibson and Crooks [15] proposed the concept of field of safe travel, in which a driver perceives a dynamic zone where a vehicle could travel safely. This concept is

popular in the literature but it has not been studied with autonomous vehicles that navigate among pedestrians. Safety margins are complicated in the analysis of mobility in shared spaces. Traffic behaviour around the vehicle is unpredictable, interactions may be more frequent, and if pedestrian density is high, then the passenger may be forced to tolerate smaller safety distances than in traditional urban environments.

The approach of an obstacle, such as a pedestrian, is specified to the observer by an increase in the optical angle subtended by the obstacle. Lee [16] demonstrates that the optical angle's rate of expansion is inversely proportional to the time to collision (TTC) if the current approach velocity–the time remaining until the obstacle is reached–is maintained. Bootsma and Craig [17] find evidence of a relationship between the TTC and the driver's perception of an upcoming collision. Following this work, other researchers have used the TTC for collision detection [18, 19]. This indicator is currently used by the automotive industry to determine alert thresholds for collision warning systems [20]. However, the TTC is only applicable when an obstacle is already in a vehicle's path. Other indicators must be computed to study the collision risk perception for more complex path intersections, such as between an autonomous vehicle and a pedestrian in a shared space. The collision risk perception from the perspective of the self-driving car's passenger may arise as soon as a pedestrian attempts to cross the vehicle's path. In these situations, one can assume that passengers will attempt to estimate the potential collision point and judge whether the vehicle is behaving correctly. Cutting et al. [21] posits that an individual can anticipate a collision based on the evolution of the angle between the direction of the vehicle's motion and the direction of an approaching pedestrian. This angle is called the *bearing angle* by Ondrej et al. [22]. A collision is predicted when the time derivative of the bearing angle is zero. The time remaining before this collision will occur can be assessed by the time to interaction (TTI), which is a generalisation of the TTC to converging trajectories in lateral and longitudinal dimensions. While the TTC can only be computed when the pedestrian is in the observer's path, the TTI can be defined independently of the pedestrian's position and relative speed, even when the probability of a collision is zero (e.g., when the trajectories are not convergent). Based on these considerations, manipulating the TTI is ideal for studying a passenger's risk perception.

## 1.3. Risk perception measurements

This study proposes an experiment based on two distinct measurements of a passenger's risk perception associated with an approaching pedestrian: the subjective risk assessment and the electrodermal activity (EDA).

The subjective assessments can be performed in real time using an analogue device, such as a rotating potentiometer. The aim is to obtain collision risk assessments without interrupting the scenarios presented to the experiment's participants. This method is similar to that used in studies regarding the perceived comfort of autonomous vehicle passengers conducted by Hartwich et al. [23] and Telpaz et al. [24]. Walker et al. [25] and Petit et al. [26, 27] confirm that comparable devices are relevant for real-time assessments of interactions between pedestrians and autonomous vehicles.

EDA corresponds to electrical variations in the skin that occur relative to sweat gland functioning under the control of the sympathetic nervous system [28]. EDA is relevant for studying emotional and cognitive states [29] and is not influenced by the parasympathetic nervous system. It therefore has the advantage of being more easily interpretable than other physiological variables, such as heart rate, ventilation or body temperature [30, 31]. The most common method for studying EDA is by measuring electrical conductance at the skin surface. Measurements are taken in micro-siemens and comprise the superposition of two distinct elements:

the tonic and phasic components. The tonic component is associated with an individual's baseline level of skin conductance, which reveals slow variations, whereas the phasic component generally indicates rapid changes in skin conductance, which are often referred to as *skin conductance responses* (SCRs). SCRs have been used as indicators of events that cause discomfort in drivers [13] or stress [31, 32]. Choi et al. [33] consider that an individual's risk perception could lead to substantial changes in EDA and that, consequently, this measure could be an adequate indicator of risk perception.

This dual-measurement approach of recording subjective assessments and physiological responses simultaneously is part of a broader issue concerning the distinction between fast, automatic and unconscious cognitive processes (Type 1) and slow, laborious and conscious ones (Type 2) [34, 35]. Subjective risk assessments belong to Type 2 processes, which rely on working memory and involve mental simulations of future possibilities to make explicit judgements. Conversely, physiological reactions belong to Type 1 processes, which are autonomous, do not require working memory and underlie implicit information processing. However, as Evans [34] notes, the nature of the distinction between the two types of processes and their mutual relationship is not unequivocal in the literature. Petit et al. [27] investigated the relationship between subjective collision risk assessments and SCRs in passengers of a simulated self-driving car that avoids pedestrians in a shared space. They showed that the reduction of safety margins increased risk perception according to both types of indicators. However, subjective assessments were more sensitive to low-risk situations than are physiological responses. Thus, declarative and physiological measures should be considered complementary rather than redundant. The main limitation of Petit et al.'s [27] study is that only two factors are manipulated to vary vehicle–pedestrian interactions. With more complex and realistic situations, new relationships between vehicle–pedestrian dynamics and the two perceptual systems may be revealed.

### 1.4. Research aims

Two main objectives guided the development of this study:

- To analyse the influence of autonomous vehicle–pedestrian crossing dynamics on passengers' collision risk perceptions;

- To investigate the relationship between a subjective measure of collision risk and the passengers' SCRs during interactions with pedestrians.

To meet these objectives, a driving simulator experiment was designed to confront participants with the situation of crossing a shared space with no priority rules between the vehicle and pedestrians. With pedestrians free to cross the street as they wished, participants in the virtual vehicle were asked to monitor the situation and assess the risk of collision with a pedestrian at any time. Seven bi-modal factors were used to manipulate the dynamics of vehicle–pedestrian interactions. These factors were chosen to vary each pedestrian's position and relative speed, initial orientation, approach angle and minimum safe distance from the vehicle. It was hypothesised that an increase in crossing speed combined with a reduction in safety margin could increase a passenger's collision risk perception. Additionally, it was hypothesised that the pedestrian's angle of approach could alter this perception.

The influence of the seven factors and the relationship between the subjective and physiological measures were analysed using hybrid Bayesian networks. This method was chosen to simultaneously answer two questions. First, which factors of vehicle–pedestrian interactions significantly influence risk perception? Second, what is the relationship between subjective risk assessments and physiological responses? Answers to these questions were achieved by

testing the likelihood of a relationship between the two types of measures under the influence of vehicle–pedestrian dynamics.

## 2. Methods

### 2.1. Participants

For this experiment, 27 volunteer participants aged 19 to 63 years ($M$ = 29.2, SD = 11.6) were recruited in May-June 2021. They comprised six females and 21 males, all of whom were licensed drivers. The only requirement for recruitment was that the participants had good eyesight, with or without correction. The experiment was approved by the non-interventional research ethics committee of Nantes University (CERNI, IRB #IORG0011023; approbation #10032021). All participants gave written informed consent in accordance with the Declaration of Helsinki. Each participant's data set was associated with an alphanumeric identifier to make the analyses anonymous.

### 2.2. Experimental setup

The urban environment was modelled on a fixed-base driving simulator (Fig 1A), and the experimental scenarios were developed with SCANeR Studio v1.9 software (AVSimulation, France).

The EDA of the participants was measured to quantify their physiological responses during the simulation. Therefore, an exosomatic recording was performed to measure skin conductance using two electrodes (DC recording method; see Caruelle et al., 2019 [36]). As illustrated in Fig 1C, the electrodes were placed on the phalanges of the participants' index and middle fingers on their non-dominant hand. To improve electrical conductivity with the skin, the electrodes were covered with isotonic gel. No skin preparation was performed prior to

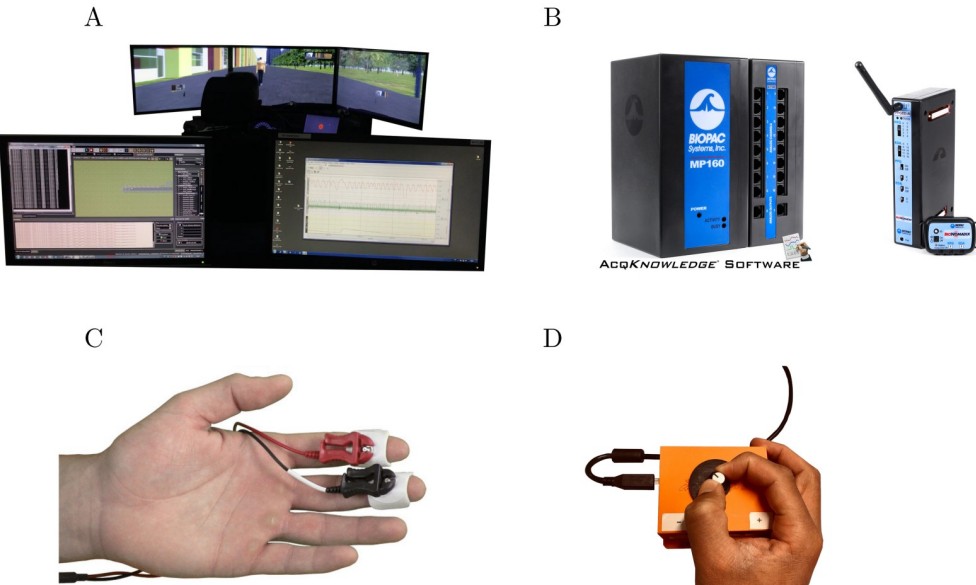

A

B

C

D

**Fig 1. Experimental setup for risk measurement.** Note. Materials for the experiment. (A) The fixed-base driving simulator. (B) The MP160 module and wireless BioNomadix device for EDA recording. (C) The two electrodes on the distal phalanges for EDA recordings. (D) The analogue device for subjective risk assessment.

electrode placement. Data were collected at 1000 Hz using a BioNomadix wireless transmitter coupled with an MP160 acquisition module (Fig 1B, BIOPAC Systems, Inc., USA).

To provide participants with the ability to assess perceived risk during the simulation, an analogue device was developed for one-handed operation. This device is presented in Fig 1D. It is a rotary potentiometer with a low stop at the extreme left (corresponding to the absence of perceived risk) and a high stop at the extreme left (corresponding to a maximum perceived risk). The amplitude of the movement is therefore 180˚. The potentiometer was connected to an Arduino Uno board integrated in a rigid plastic case that was custom designed and 3D printed. The device was conceived in such a way that it would not cause visual distraction or require participants to look at it. The device was placed on each participant's lap to be manipulated by the dominant hand (i.e., the hand without the skin conductance electrodes). Subjective risk assessment data were collected at 20 Hz.

## 2.3. Design of experiment

In this experiment, participants were seated in a vehicle traveling along a straight path in an urban street shared with pedestrians. Pedestrians were programmed to cross the street in different ways depending on seven bi-modal manipulated factors. The factors and their modalities are presented in Fig 2 and detailed in Table 1. Manipulating the pedestrians made it possible to control three main aspects of vehicle–pedestrian interaction. First, two factors defined the initial condition of the pedestrian on the lane relative to the vehicle: *Side* and *Orientation*. Second, three factors determined the relative motion of the pedestrian and the vehicle: *v-Speed*, *p-Speed* and *Angle*. Finally, two factors determined the relative position of the

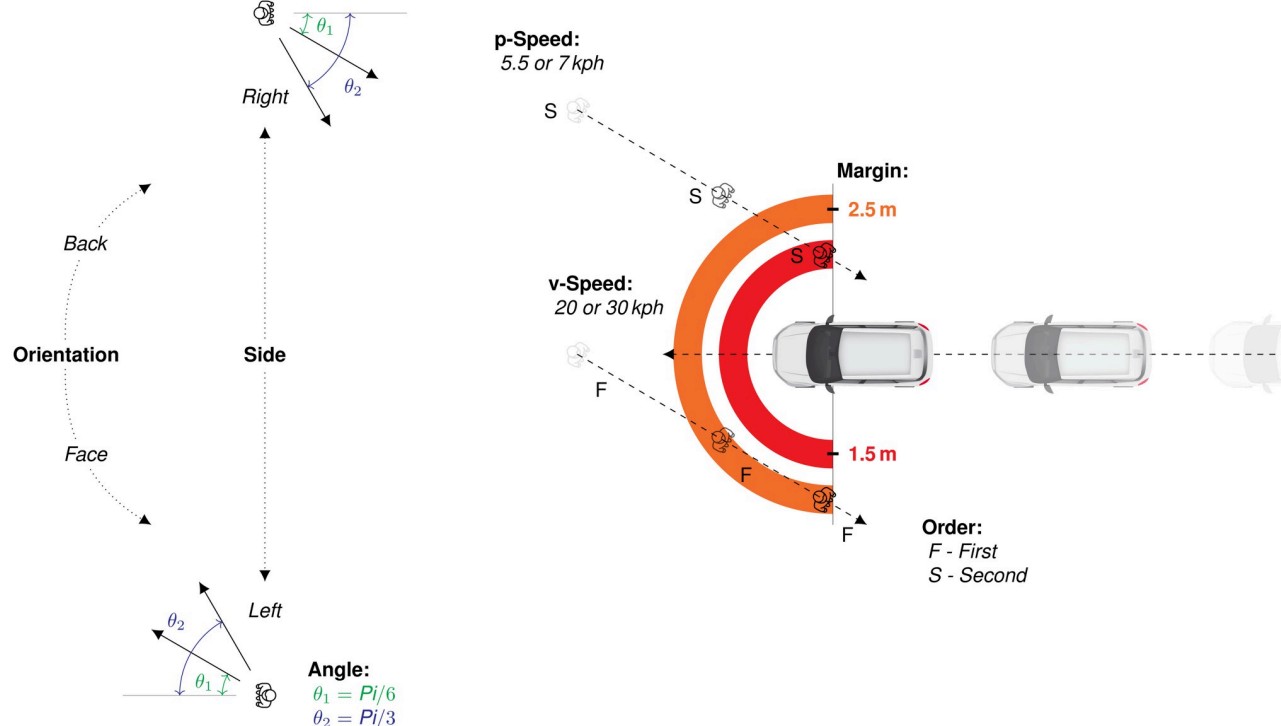

**Fig 2. Schematic representation of the seven manipulated factors.** Note. For the purpose of illustration, several pedestrians are presented simultaneously, although they were spaced approximately 20 s apart in the experiment. Additionally, the distances between the pedestrians and the vehicle are not scaled.

**Table 1. Description of the seven manipulated factors.**

| Factor | Detail | Levels |
|---|---|---|
| *Initial conditions of the pedestrian* | | |
| Side | Initial pedestrian position on the street, relative to the vehicle | Left, Right |
| Orientation | Initial pedestrian orientation, facing or turning its back to the vehicle | Face, Back |
| *Relative motion* | | |
| v-Speed | Autonomous vehicle speed | 20 kph, 30 kph |
| p-Speed | Pedestrian walking speed | 5.5 kph, 7 kph |
| Angle | Crossing angle of the pedestrian, relative to the street | $\pi/6$, $\pi/3$ |
| *Relative positions at close proximity* | | |
| Order | Order of passage of the pedestrian | First, Second |
| Margin | Closest distance between the pedestrian and the vehicle | 1.5 m, 2.5 m |

pedestrian and the vehicle in close proximity: *Order* and *Margin*. For the latter factor, the distance was calculated relative to the driver's seat, that is, the participant's position in the driving simulator.

A full factorial design involving these seven factors would have resulted in $2^7$ = 128 crossing situations. This experimental design was not appropriate because exposing participants to a lengthy experiment could have created habituation to crossing situations and caused a decrease in the intensity of the participants' responses. The combination of the seven factors was therefore established according to a $2^{(7-2)}$ fractional design ([37], Chapter 6). Experimental design constructed in this way is balanced up to third-order interactions.

This experimental design was submitted to each participant (within-subject design). The initial experimental design was randomised so that each participant encountered the 32 situations in a different order. To ensure that participants were indeed assessing a risk of collision, and to prevent overconfidence in the system, four additional situations were added in which the trajectories of the pedestrian and the vehicle intersected. These conditions were introduced at ranks 7, 14, 21, and 28. In these cases, p-speed was set to 7.5 kph and v-speed to 30 kph. When the collision occurred, the vehicle passed through the pedestrian without any other visual effect. As expected, these four trials resulted in a very high perceived risk, both subjectively and physiologically. This trivial result will not be detailed, but it suggests that participants were well aware that the risk of colliding with a pedestrian existed.

In summary, the participants experienced 36 crossing situations. Each situation was different, and four were collisions. The pedestrians were separated by approximately 20 s. The entire scenario lasted 18 min 30 s.

The simulated pedestrians were all adult males. To reduce the predictability of the crossing situations, 72 non-crossing pedestrians were placed on each side of the lane. Their lateral position in the lane and walking speed varied.

## 2.4. Dependent indicators

Two indicators were calculated to quantify the participants' risk assessments. For each crossing situation, the area under the curve of assessed risk was calculated to constitute the indicator *iSA*. This value reflected the dynamics of the risk assessments. The maximum value of the assessed risk, denoted *mSA*, was also calculated. The values of these two indicators were calculated for each crossing situation. An example of these indicators is presented in Fig 3B. They were calculated using the R software [38].

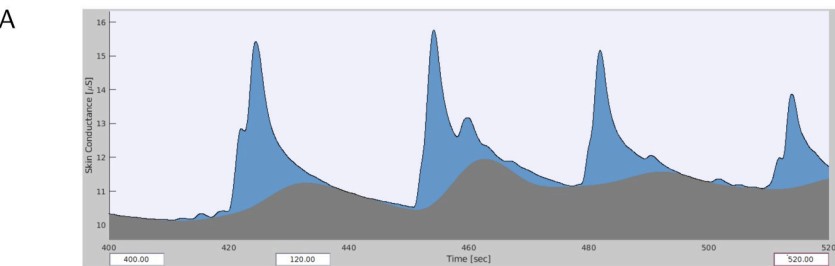

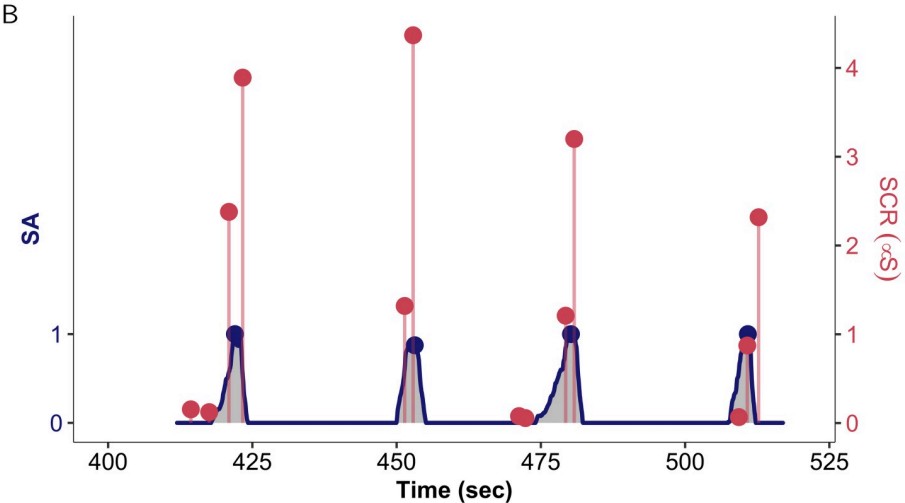

**Fig 3. Samples of subjective assessment and EDA data.** Note. Samples of subjective assessment (SA) and EDA recordings for a participant during four successive crossing situations. (A) Screenshot of the data presented in the Ledalab framework. The tonic component is represented by the grey area. The phasic component is represented by the blue area. (B) A sequence of processed data where SA and SCR indicators are represented together. The SA series is indicated by the blue line. For each situation, the value of the iSA corresponds to the grey area below the line, and the value of the mSA is indicated by a large blue dot. The occurrences of SCRs are represented by vertical red segments whose height corresponds to the response amplitude. For each situation, the value of nSCRs corresponds to the number of responses (i.e., four for the first pedestrian, two for the second, etc.), and the value of the mSCRs is indicated by a large red dot.

Two indicators were calculated from the participants' SCRs: the number of SCRs, denoted *nSCR*, and the maximum amplitude of these responses, noted *mSCR*. An example of these indicators is also provided in Fig 3B. The data were initially processed using AcqKnowledge 5.0 software (BIOPAC Systems, Inc., USA). Matlab 2018 and R were then used to extract the indicators.

The calculation of the EDA indicators was similar to that performed by Petit et al. [27]. Several manipulations were performed to calculate the indicators. First, the raw data were preprocessed using AcqKnowledge software, following the recommendations of Braithwaite et al. [30] and Findlay [39]. This included resampling at 50 Hz, moving median smoothing per 1 s period and performing low-pass filtering at 1 Hz. The preprocessed data were then analysed using the Ledalab v3.4.9 application. The SCRs were extracted from the phasic component of each EDA, which was identified by the continuous decomposition analysis algorithm (CDA) [40]. Fig 3A illustrates an example of the result of the CDA algorithm over data from four crossing situations. Only responses with an amplitude greater than 5 µS were retained.

Finally, participants' SCRs were recorded for each crossing situation as soon as the pedestrian began to cross the lane. Moreover, to account for the delay between a stimulus and the corresponding SCR, the related SCRs for each crossing situation were considered up to 3 s after the pedestrian passed and was moving away from the vehicle [41, 42].

## 2.5. Procedure

When each participant arrived, the purpose of the experiment and its general course were presented. Participants were asked to read and sign an informed consent document. Subsequently, skin conductance electrodes were placed on each participant's non-dominant hand. A test measurement was then conducted to ensure that the participant was responsive ([41], p. 439).

The participant was then invited to sit in the driving simulator and learn to use the subjective assessment device. Instructions were offered regarding the rotating potentiometer. A test simulation was presented to each participant. This simulation consisted of the same virtual environment as the experimental trials except that no pedestrians were crossing the road. During this training, the participant received visual feedback regarding the position of the rotating potentiometer on the central monitor (see Fig 4). Participants were asked to reach 33%, 50% and 66% of the maximum position of the potentiometer with their eyes closed, that is, relative to the upper stop. They had to be able to reach these positions starting from the minimum position (low stop) and then starting from the maximum position (high stop). The training ended as soon as the participant completed the objectives and felt sufficiently accustomed to the subjective assessment device. Each participant was informed that visual feedback of the potentiometer's position would not be available for the remainder of the experiment. Next, participants were presented with three different crossing situations to get familiar with the upcoming task.

Following this training, participants were told that they would experience 36 crossing situations with different safety margins. The experimenter specified that in some situations, collisions would occur. Participants were reminded of the experiment's instructions. They were asked to assess the collision risk throughout the vehicle's travel without interruption.

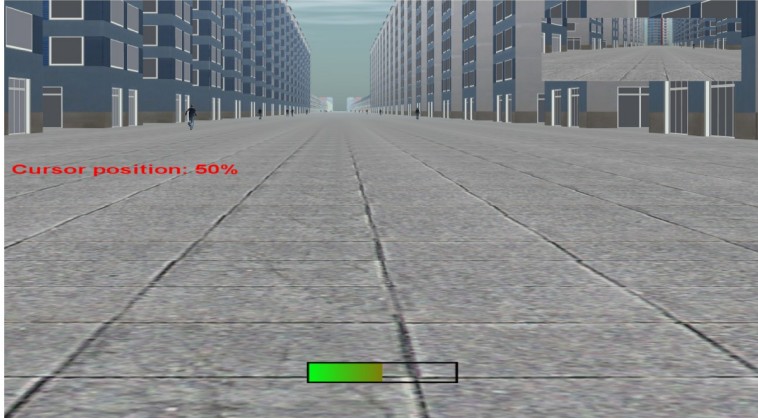

**Fig 4. Screenshot of the centre monitor during the initiation phase.** Note. During the training phase, no pedestrian crossed the lane. Two instances of visual feedback of the potentiometer position were displayed: a text specifying the cursor position as a percentage of the maximum value and a coloured horizontal bar at the bottom of the centre monitor.

Participant were further told that the potentiometer should reflect their perception of the collision risk as accurately as possible in real time. Specifically, they were to remember to return to zero (i.e., the low stop) as soon as they assessed no collision risk. Additionally, the experimenter specified that they could remain in the low stop position for the entirety of a crossing situation if they did not perceive any risk.

Finally, the experimental simulation was initiated when participants stated their readiness. The participants' view during the lowest safety margin crossing situations is provided in Fig 5.

The purpose of this study was to analyse the effect of different factors of vehicle–pedestrian crossing dynamics on assessed risk and to analyse the relationship between physiological and subjective measures.

To achieve these two objectives, the relationships between the factors and the calculated indicators were modelled via Bayesian networks. This method, based on stochastic distributions, allows to discover the best structure of relationships (i.e., the one that best fits the data) between manipulated factors and the dependent measures. Bayesian networks are ideal for taking an event that occurred (for example, a behavioural or physiological response) and predicting the likelihood that any one of several possible known causes was the contributing factor. A Bayesian network represents a set of variables, called *nodes*, and their conditional dependencies via a directed acyclic graph. A relationship between two nodes, that is, between a factor and an indicator or between two indicators, is represented by a directional arrow. In this study, the indicators were analysed in pairs: (iSA, nSCR) and (mSA, mSCR). This pairwise grouping was chosen to examine the relationships between indicators of the same nature, that is, indicators related to the dynamics of a measurement and indicators related to the maximum amplitude of measurements, respectively.

As illustrated in Fig 6, the directed acyclic graphs studied each had nine nodes: One node was assigned to each of the seven factors, another was assigned to the subjective assessment indicators and the ninth was assigned to the SCR indicators.

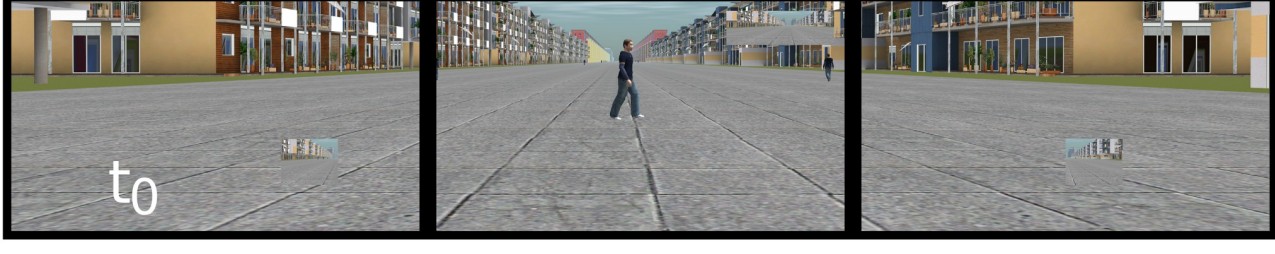

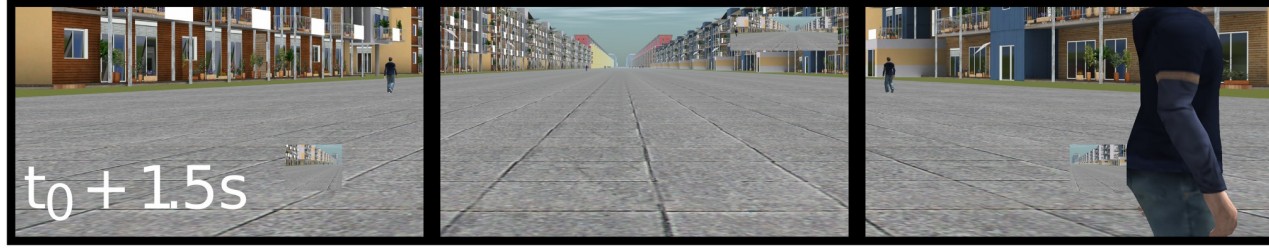

**Fig 5. Screenshots of the three monitors during the experiment.** Note. The two screenshots illustrate participants' view during a crossing situation at a time interval of 1.5 s. In this situation, the vehicle was driving 20 kph (vehicle speed = 20 kph). The pedestrian was crossing the street from left to right (pedestrian position = left) at 5.5 kph (pedestrian speed = 5.5 kph) facing the vehicle (orientation = face) with an angle of Pi/3 (angle = Pi/3). The pedestrian passed in front the vehicle (order = first), respecting a safety margin of 1.5 m (margin = 1.5 m).

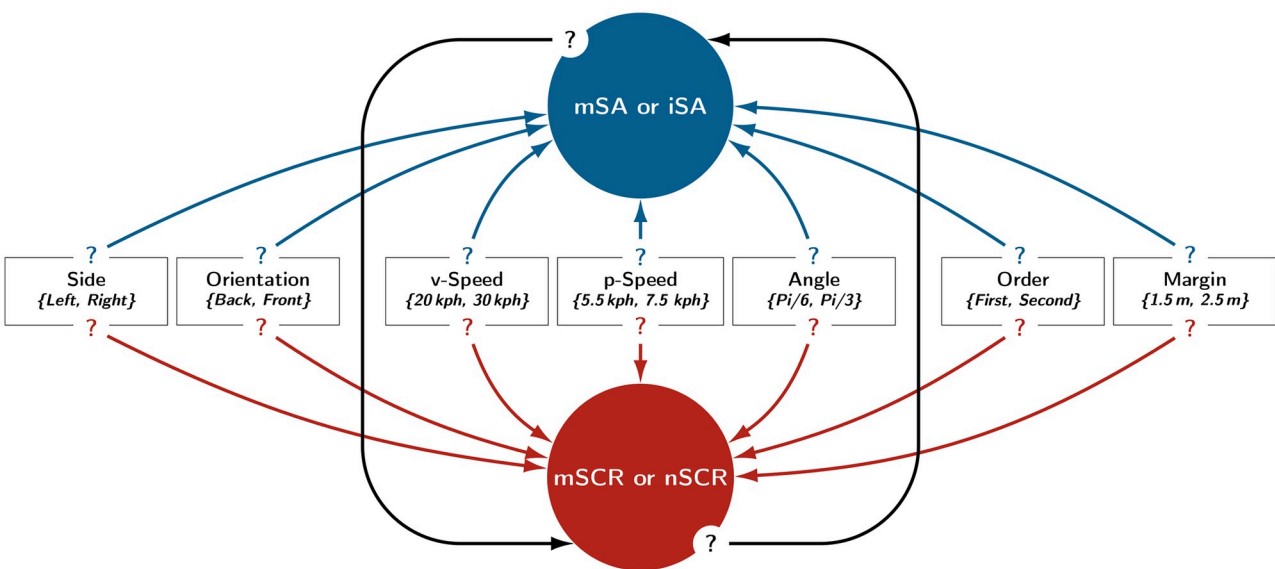

**Fig 6. Directed acyclic graphs with potential relationships.** Note. This figure represents all the potential relationships that can exist in the Bayesian networks evaluated in this experiment. Potential relationships between experimental factors and indicators are represented by arrows with question marks. A total of 12,288 distinct Bayesian networks were compared for each of the two pairs of indicators: (iSA, nSCR) and (mSA, mSCR).

For each pair of indicators, there are 12,288 different Bayesian networks according to the following calculation:

$$3 \times \left( \sum_{0 \leq i \leq 3} \binom{7}{i} \right)^2 = 12288 \tag{1}$$

where $\binom{7}{i} = \frac{7!}{i!(7-i)!}$ denotes the number of combinations that can be represented by an indicator in relation to exactly $i$ distinct factors. This calculation incorporates the following conditions:

1. The factors cannot be related to each other because they correspond to independent variables that are manipulated in the experimental design;

2. An indicator can only be related to a maximum of three factors simultaneously.

The second condition is the consequence of the experimental design. By its construction, some fourth-order interactions between factors cannot be analysed with balanced sample sizes. This is not a concern since such complex interactions would be difficult to interpret.

A comparison algorithm was developed to compare these Bayesian networks according to the Bayesian information criterion (BIC; [43, 44]), which is based on the likelihood of the data obtained during the experiment. This likelihood is itself based on the estimation of the parameters of the statistical distributions chosen for each of the nodes [27, 45].

In Bayesian networks, a probability distribution must be assigned to each variable (factor or dependent variable). In this case, the factors have been treated as bimodal random variables, since the factor modalities appear randomly from the participants' point of view. According to the experimental design, the two-level distributions of each factor were balanced. Therefore, the probability mass functions of all independent factors were equal. For any factor, the probability of observation of its level was 1/2.

Two specific distributions were assigned to the nodes that corresponded to the subjective assessment indicators and the SCR indicators, as detailed below.

Subjective evaluation indicators followed Gaussian distributions and required some preprocessing. First, a skewness correction was performed for each indicator: values of the indicator iSA were raised to the power 1/2, and the values of the indicator mSA were raised to the condition of 1/3 power. Then, to reduce inter-participant variability and to model all data simultaneously, the indicator values were centred and reduced by participant. S1 Table presents the effect of the transformations on the mean, standard deviation, skewness and kurtosis values of each indicator.

The SCR indicators followed a compound Poisson–gamma distribution, which have positive mass at zero (participants had no skin response in some situations), but are otherwise positive and continuous. Thus, a Tweedie distribution was well suited to model the SCRs. The Tweedie distribution is characterised by three parameters and is denoted $Tw_p$. The parameter $\mu$ is the mean of the distribution, $\varphi > 0$ is the dispersion parameter and $p$ is used to express the relationship between the variance and the mean of the distribution such that the variance is equal to $\varphi\mu^p$, with $1 < p < 2$. SCR indicators were also standardised in order to reduce inter-participant variability.

A random variable $Y$ following a $Tw_p$ distribution can be presented as

$$Y = \sum_{i=0}^{N} X_i \tag{2}$$

where $N$ is a Poisson random variable and the $X_i$ are independently, identically distributed random variables of Gamma distribution. According to Dunn and Smyth [46], it can be demonstrated that

$$Pr(Y = 0) = exp\left(\frac{\mu^{2-p}}{\varphi(2-p)}\right). \tag{3}$$

Eq 3 was used in the analysis to obtain an estimate of the probability that a SCR indicator (nSCR or mSCR) was equal to 0.

Parameter estimates for the distributions were made using R software. For the subjective assessment indicators, Gaussian distribution parameters were estimated with the *stats* package (R Core Team, 2020); these parameters corresponded to maximum likelihood estimates. For the EDA indicators, the parameters of the Tweedie distributions were estimated with the methods developed in the packages *tweedie* [46–48] and *cplm* [49].

The models were ranked in ascending order according to their BIC scores, with the best model having the lowest score. Raftery's [43] comparison thresholds were used to determine the degree of evidence for the relationship of one Bayesian network compared to another. These thresholds are detailed in Table 2 and provide guidelines for comparing Bayesian networks with each other and for discussing the significance of relationships between network nodes (factors and indicators).

The best Bayesian networks were finally detailed to examine the nature of the relationships.

For subjective assessment indicators, a cluster analysis [50, 51] was performed on the estimated means of the Gaussian distributions. A cluster analysis allows researchers to compare non-nested models based on the BIC. It was implemented using the R package *partitions* [52, 53]. This method enables scientists to evaluate the degree of homogeneity of the Gaussian distributions in the best Bayesian networks. The primary purpose of the cluster analysis was to objectively group experimental conditions that produced an equivalent level of risk perception. A secondary, more exploratory objective was to determine whether a specific level of risk was

**Table 2. Grades of evidence corresponding to BIC differences between two Bayesian networks.**

| BIC Difference | Evidence |
|---|---|
| [0; 2] | Weak |
| ]2; 6] | Positive |
| ]6; 10] | Strong |
| > 10 | Very Strong |

Note. Bayesian networks were ranked by their BIC scores. This table (from Raftery, 1995, p. 139) indicates thresholds that were used to discuss the degree of evidence for differences between networks.

due to homogeneous causes. For instance, a high level of risk perception could correspond to either a set of conditions associated with a particular degree of a factor or with the interaction between two factors.

For SCR indicators, the estimates of the mean were plotted, as were the probability of zeros (when no SCR occurred). These two representations provided a comprehensive overview of the Tweedie distributions obtained from the estimated parameters ($p$, $\mu$, $\varphi$). Specifically, these representations permitted to place the evolution of the mean of an indicator parallel to the probability of non-response (see Eq 3).

## 3. Results

The best Bayesian networks, according to the BIC [44] were selected only if they obtained a BIC that was at least 4 points lower than the others. Such a difference corresponds to a positive or stronger degree of evidence, according to the Raftery thresholds (see Table 2).

For the iSA and nSCR indicators, the best network obtained a BIC that was 4.6 points better than the second-best network. Thus, only the first network will be considered (Fig 7). For the

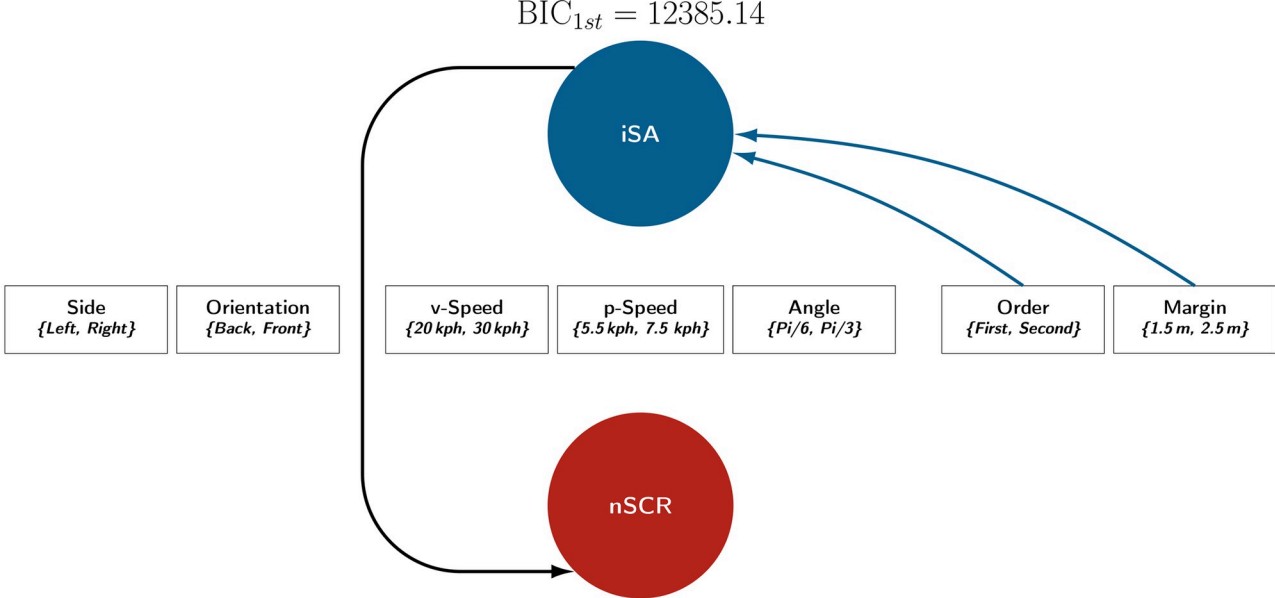

**Fig 7. Best Bayesian network for the indicators iSA and nSCR.** Note. This figure presents the best Bayesian network (of the 12,288 computed networks) related to the indicators iSA and nSCR. This Bayesian network presented the lowest BIC score with a positive grade of evidence (Raftery, 1995) compared to the second-best network.

mSA and mSCR indicators, the best network obtained a BIC that was 1.6 points better than the second best and 4 points better than the third best. These two best networks are presented in Fig 9. In both Figs 7 and 9, the BIC of the first network is denoted $BIC_{1st}$. Details of the distribution estimates are provided in S1 and S2 Figs. To provide an additional overview of the factor effect, the marginal means for each level are presented in S2 Table.

### 3.1. The best Bayesian network for the indicators iSA and nSCR

Fig 7 portrays the relationships in the best Bayesian network obtained for the pair of indicators related to the dynamics of subjective assessments and SCRs.

This Bayesian network reveals that only two of the seven factors influenced the indicator iSA: the pedestrian's order of passage when crossing with the vehicle and the safety margin between the pedestrian and the vehicle.

According to the means presented in Fig 8A, iSA values were greatest when pedestrians passed in front of the vehicle (first) rather than behind the vehicle (second). Additionally, this graph presents higher iSA values for the smallest safety margin, which is 1.5 m between the vehicle and the pedestrian. For the smallest safety margin, the increase in means between the two levels of the factor Order is 0.98. This increase is equal to 0.72 for the largest safety margin (2.5 m), which is of the same order of magnitude. This indicates a weak interaction between the two factors.

Additionally, the cluster analysis revealed that the highest level of risk perception (triangle pointing up in Fig 8A) was observed when the pedestrian passed in front of the vehicle with a

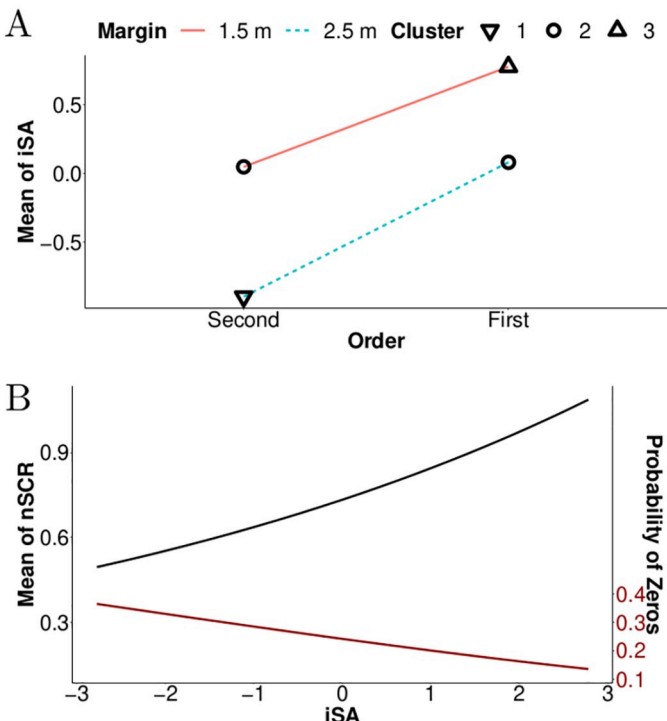

**Fig 8. Results of the best Bayesian networks concerning means of the indicators iSA and nSCR.** Note. (A) The means of the indicator iSA as a function of Margin and Order. (B) The estimated mean of the nSCR and the probability of zero response as a function of the indicator iSA (normalised values). Results are reported based on relationships elicited in the best Bayesian network (see Fig 7). The three symbols (triangle pointing up, circles, and triangle pointing down) correspond to distinct levels of perceived risk as revealed by the cluster analysis.

1.5 m safety margin. Conversely, the lowest level of risk perception (triangle pointing down) was observed when the pedestrian passed behind the vehicle with a 2.5 m safety margin. The two other conditions (circles) produced an intermediate level of subjective risk.

Five of the seven manipulated factors do not appear in the best Bayesian network: Side, Orientation, v-Speed, p-Speed and Angle. These factors are related to the initial conditions and the dynamics of the crossing situations, which indicates that the subjective estimation of the collision risk was mainly influenced by the factors that determined the relative positions of the pedestrian and the vehicle when the two were near each other.

Furthermore, the Bayesian network presented in Fig 7 suggests that the indicator iSA is implicated in the coefficient estimation of the Tweedie distribution for the indicator nSCR, which suggests that variations in subjective assessments partially explain the nSCRs. Fig 8B illustrates that the increase of iSA causes an increase in nSCR and decreases the probability of non-response. The nSCR is therefore better explained by subjective risk assessments rather than by the external factors manipulated in this experiment.

## 3.2. The best Bayesian networks for the indicators mSA and mSCR

Fig 9 presents the two best Bayesian networks obtained for the indicators mSA and mSCR. These two networks are identical except for the relationship between the factor Orientation and the indicator mSA, which only appears in the second-best network. This relationship penalises the BIC score by 1.6 points. Thus, both networks differ with a low degree of evidence, according to Raftery's criteria. The factors Margin and Order had the greatest influence on the indicators. These are the only two factors included by the best Bayesian network (Fig 9A). However, results demonstrate that the factor Orientation modulated the values of the indicator mSA. This factor appears in the second-best Bayesian network (Fig 9B).

Fig 10A portrays the effect of factors Margin, Order and Orientation on the mean of the indicator mSA. A smaller safety margin increased mSA values. When the pedestrian crossed in front of the vehicle, mSA values were also higher. These two effects appear to be cumulative. However, the influence of the factor Order was modulated by the factor Orientation: when pedestrians crossed the street moving in the same direction as the vehicle, the effect of the factor Order was larger than when the pedestrians were facing the vehicle. These increases were similar for both safety margins tested. On average, mSA values increased by 1 when pedestrians moved in the same direction as the vehicle and passed in front rather than behind. The average increase was 0.23 when pedestrians crossed the street facing the vehicle. Thus, participants perceived a greater difference when pedestrians were moving in the same direction as the vehicle.

The cluster analysis distinguished five levels of mSA values, as indicated by the symbols in Fig 10A. The two highest subjective risk levels (triangle pointing up and squares) each corresponded to conditions with a safety margin of 1.5 m when pedestrians were facing the vehicle (regardless of the crossing order) or when they turned their back to the vehicle while passing first. The two lowest subjective risk levels (triangle pointing down and diamond) each corresponded to conditions in which the pedestrian passed second with a safety margin of 2.5 m. The final three conditions (circles) resulted in an intermediate mSA level.

These two best Bayesian networks are consistent, as both demonstrate the relationship between the indicators mSA and mSCR. Fig 10B reveals the evolution of the indicator mSCR as a function of the mSA values. The figure is similar to that presented for the indicator nSCR; however, the mean of the indicator mSCR increases proportionally faster as a function of the mSA values. The exponential power coefficient is 0.2 for the indicator mSA while it is 0.14 for the indicator iSA (see Figs 8B and 10B).

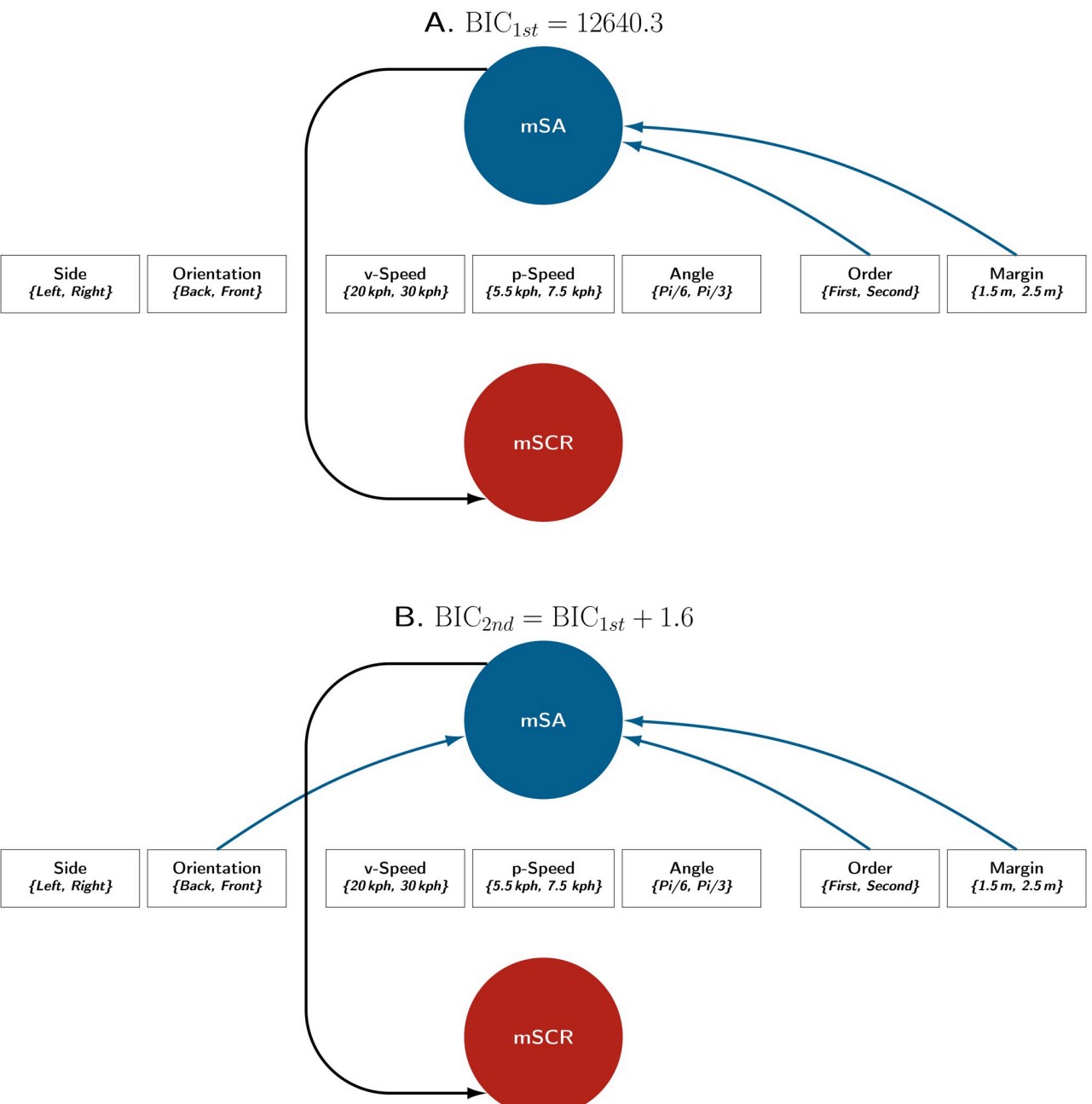

**Fig 9. Best Bayesian networks for the indicators mSA and mSCR.** Note. These are the two best Bayesian networks involving the mSA and mSCR indicators (of the 12,288 computed networks). These two networks presented the lowest BIC scores with at least a positive grade of evidence (Raftery, 1995) when compared to other networks. However, the difference between them corresponded to a low grade of evidence.

In conclusion, the results demonstrate a relationship between the indicators of the two risk perception measures: indicators iSA and nSCR on the one hand and indicators mSA and mSCR on the other. Moreover, only three of the seven factors were retained by the best Bayesian networks. The speeds of the vehicle and the pedestrian as well as the side position of the pedestrian and the angle of incidence of its trajectory did not significantly affect the indicators.

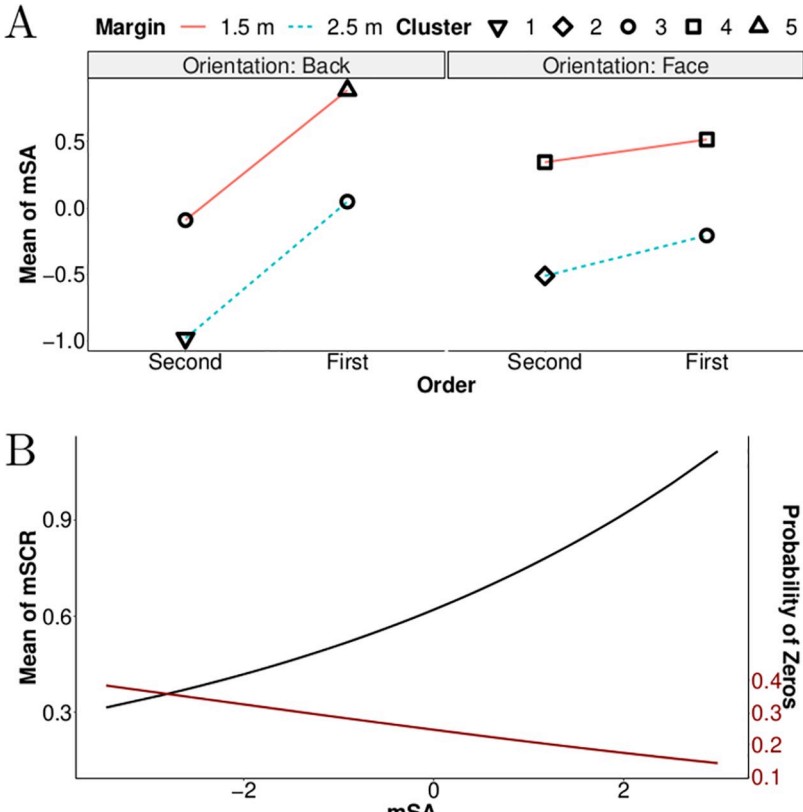

**Fig 10. Results of the best Bayesian networks concerning means of the indicators mSA and mSCR.** Note. (A) The means of the indicator mSA as a function of Margin, Order and Orientation. (B) The estimated mean of mSCR and the probability of zero response as a function of the indicator mSA (normalised values). Results are reported based on relationships elicited in the best Bayesian networks (see Fig 9). The five symbols (triangle pointing up, diamond, circles, squares and triangle pointing down) correspond to distinct levels of perceived risk as revealed by the cluster analysis.

## 4. Discussion

This study investigated how the passenger of an autonomous vehicle perceives the risk of collision with pedestrians in a shared space. A Bayesian network analysis was used to ascertain which of seven factors determine the dynamics of vehicle–pedestrian interaction and influence risk perception. The dependency between subjective risk assessments and induced SCRs was also examined.

### 4.1. Risk perception determinants in shared spaces

The best Bayesian networks indicate that the safety margin (minimal distance between the vehicle and the pedestrian) and the pedestrian's order of passage (whether the pedestrian crosses the vehicle's path before or after the vehicle passes) were the most significant determining factors in evaluating collision risk. On average, the indicators of subjective assessments were higher when the safety margin between the pedestrian and the vehicle was 1.5 m rather than 2.5 m. The averages were also higher when pedestrians passed in front of the vehicle rather than behind it. Since those effects appeared to be cumulative, the highest risk perception was observed when pedestrians passed in front of the vehicle with a margin of 1.5 m. A third

factor of secondary importance (whether the pedestrian faced or had their back to the vehicle) modulated the maximum amplitudes of participants' responses.

It was expected that risk assessments would vary as a function of the relative speed and the pedestrians' angle of approach. These factors conditioned the evolution of the bearing angle and the TTI [22]. During the crossing situations, vehicle and pedestrian speeds were constant, as was the pedestrian's crossing angle; assessing the bearing angle or TTI may have allowed participants to accurately determine whether a collision would occur [21, 22, 54]. However, results demonstrate that these factors were not decisive in the participants' assessments of collision risk. Instead, participants waited until the pedestrians were close to the vehicle to judge the imminence of a collision. Thus, spatial proximity was more important than time-dependent variables derived for visual cues. This conclusion may only be valid for situations in which pedestrians and vehicles move at low speeds, as in shared spaces.

This idea aligns with Hall's [55] work on proxemics. According to this principle, the distances that individuals maintain from the elements of their environment and the associated feelings are governed by four zones (intimate, personal, social and public). Several recent studies have proposed that the proxemic approach could be relevant in analysing the feelings of a passenger in a self-driving car [56, 57]. Only when interactions intervene in a passenger's personal zone does the passenger begin to perceive a collision risk. Generally, the passenger could define a dynamic personal space in which any irruption could cause the perception of a collision risk. In the context of shared spaces, situations in which the autonomous vehicle would necessarily come close to a pedestrian could be frequent. In these situations, the passenger's risk perception could present a limitation in the acceptance of autonomous vehicles.

## 4.2. Relationship between implicit and explicit risk perceptions

The relationship between subjective risk assessments and SCRs was examined, and results confirm those of a previous study, which investigated collision risk assessments in pedestrian avoidance manoeuvres [27]. Only two factors influenced vehicle–pedestrian dynamics in that study and the results revealed that variations in subjective risk assessments could explain the observed variations in the nSCRs and the mSCRs. The present study provides new evidence for the relationship between subjective assessments and EDA. Specifically, results demonstrate that variations in SCR indicators were more likely to be explained by variations in subjective assessments than by variations in the seven manipulated factors. Furthermore, the exponential function regarding subjective ratings and SCRs as well as the fact that less risky situations more often resulted in a lack of SCRs suggest that physiological responses reflect the perception of relatively high risk, whereas subjective assessments are more progressive.

SCRs and subjective assessments can be assumed to belong to two distinct types of risk perception: Type 1 and Type 2 processes [34, 35]. Under the control of the autonomic nervous system, SCRs associated with Type 1 processes reveal an implicit, rapid and automatic reaction of the organism to a collision hazard. Subjective assessments, associated with Type 2 processes, relate to the cognitive processes involved in the individual's explicit risk assessment [58]. The literature offers no consensus concerning the nature of the relationship between the two types of perception. This study provides a concrete example, indicating that individuals' subjective assessments of collision risk may influence their physiological states rather than the contrary.

This result may seem counterintuitive if one considers that Type 1 processes are believed to be faster than Type 2 processes to allow for a spontaneous reaction by the individual. This apparent paradox is discussed with two considerations; one is methodological, and the other relates to the context in which the risk assessments are performed. Methodologically, EDA properties presuppose that no conclusions can be drawn regarding the temporal relationships

between the two measures. Indeed, as many as 3 s can pass before an SCR is observable in a participant who has undergone a stimulus [41]. Consequently, the order in which SCRs appear relative to the subjective assessments does not play a role in the proposed models. The dependent relationship found in the best models should therefore not be considered as a temporally dependent relationship, despite what representations in the directed acyclic graphs may suggest. The modelling performed in this study is based solely on variations in the magnitude or intensity of subjective assessments and SCRs. This approach identifies the relationship between the measures independent of the influence of factors in the vehicle–pedestrian interaction dynamics. Measures of both types of risk perception may have evolved without dependent relationships; alternatively, Type 1 perception measures may have influenced the Type 2 measures. However, the opposite relationship emerged.

The relationship between the two types of risk perception processes could, however, be specific to the context of the experiment. Indeed, in this study, participants experienced a progressive risk: they had time to analyse and evaluate pedestrian interactions without any sudden danger appearing. The most likely hypothesis is that the risk was rationally assessed as long as the converging trajectories were not critical. Conversely, when the proximity to the pedestrian became excessive, the subjective perception of a collision risk increased sharply and appeared to have instigated Type 1 processes in response. The relationship between the two types of measures may have been different if a pedestrian suddenly appeared in front of the vehicle. In such cases, it would have been possible to observe a dependency relationship between SCRs and subjective assessments or even SCRs alone without variations in participants' subjective assessments, as they would not have had time to report a collision hazard on the potentiometer. Consequently, it would be interesting to conduct further studies with variations in the contexts and situations encountered by participants. In addition to relying on SCRs, future studies could also consider other measures of Type 1 processes. One could consider, for example, other physiological indicators that are influenced by the sympathetic nervous system, such as pupillary dilation or heart rate, as well as measures of participants' attitude, posture or facial expressions in a collision risk situation.

## 5. Limitations and perspectives

In this study, seven different factors were manipulated to explore a variety of conditions. The choice was made to evaluate only two degrees per variable, which limits the generalization of conclusions. For example, if higher vehicle speeds had been assessed, it is likely that this factor would have contributed more to explaining the variations in perceived risk. It should be noted, however, that the speeds evaluated in this experiment are already high in comparison with the limits generally imposed in shared spaces. Nevertheless, it would be interesting to explore in greater detail the most decisive factors.

A possible extension of this work would be to relate, for each crossing situation, the evolution over time of an objective risk indicator such as the TTI to the dynamics of subjective risk assessment (For a attempt at doing this, see [26]. A function that modulates the objective risk provided by the TTI according to the estimated subjective assessment could then be defined. In turn, this function could be integrated into trajectory planning algorithms to avoid anxiety-provoking situations for the passenger.

One way to improve passengers' collision risk estimation is to develop internal and dynamic human–machine interfaces [11] to enhance passengers' ability to predict the trajectory of incoming pedestrians and to allow them to build an accurate internal model of the vehicle dynamics through interaction. This question is still largely unexplored, and the form these interfaces might take remains a matter for speculation. It could, for example, be a progressive

change of colour on a HMI representing pedestrians perceived by the system in the vicinity, or a representation of the anticipated trajectories of the vehicle and pedestrians showing their point of intersection.

This study did not assess the influence of changes in pedestrian intention or in-vehicle behaviour. In real-life situations, vehicle speed adjustments are key in interactions with pedestrians [59], and pedestrians may ultimately decide to stop before crossing the lane in front of an autonomous vehicle [5–7]. These factors must be investigated in future work to determine their impact on passenger perceptions. It should also be noted that, while driving simulations provide better experimental control than real-life experiments, the results observed are often only of relative validity, i.e. if the effects observed are of the same nature as in real life, the values observed may differ [60]. In addition, it is difficult to account for subtle differences in the posture or facial expressions of human avatars, for example, elements that can influence vehicle-pedestrian interactions.

## 6. Conclusion

A Bayesian network analysis revealed two predominant results. Of the factors that varied in pedestrian-approach dynamics, only those that intervened when the pedestrian was close to the vehicle influenced the collision risk assessments. During crossing situations, EDA variations seemed to depend on the subjective collision risk assessments.

These results suggest that collision risk assessments typically occur in situations of close proximity to pedestrians. In other cases, the self-driving vehicle could continue to navigate among pedestrians without causing any stress to its passenger. This raises the question of the safety margins that an autonomous vehicle's passenger may tolerate when moving in a shared space. The definition of such safety margins in dynamic pedestrian interactions will be the subject of future work. Additionally, the results of this study suggest that cognitive processes involved in collision risk assessments may lead to physiological adaptations. Further studies should investigate whether this relationship can be generalised to other physiological measures.

## Supporting information

**S1 Table. Descriptive statistics of the indicator distributions.**
(DOCX)

**S2 Table. Indicator means as function of the seven independent factors.**
(DOCX)

**S1 Fig. Best Bayesian network with detailed distributions for iSA and nSCR indicators.**
(DOCX)

**S2 Fig. Best Bayesian network with detailed distributions for mSA and mSCR indicators.**
(DOCX)

## Author Contributions

**Conceptualization:** Jeffery Petit, Camilo Charron, Franck Mars.

**Data curation:** Jeffery Petit.

**Formal analysis:** Jeffery Petit, Camilo Charron.

**Funding acquisition:** Franck Mars.

**Investigation:** Jeffery Petit.

**Methodology:** Jeffery Petit, Camilo Charron, Franck Mars.

**Project administration:** Franck Mars.

**Resources:** Franck Mars.

**Software:** Jeffery Petit.

**Supervision:** Camilo Charron, Franck Mars.

**Validation:** Camilo Charron, Franck Mars.

**Visualization:** Jeffery Petit.

**Writing – original draft:** Jeffery Petit.

**Writing – review & editing:** Camilo Charron, Franck Mars.

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
