## [Decision Letter · Decision Letter 0]

9 May 2023

PONE-D-23-10461Risk Assessment and Associated Electrodermal Activity of a Self-Driving Car Passenger in an Urban Shared SpacePLOS ONE

Dear Dr. Mars,

Thank you for submitting your manuscript to PLOS ONE. After careful consideration, we feel that it has merit but does not fully meet PLOS ONE’s publication criteria as it currently stands. Therefore, we invite you to submit a revised version of the manuscript that addresses the points raised during the review process. Please consider the reviewers' comments carefully, in particular, suggestions on how to improve the description of the experimental setup and the analysis methods used. Neither the reviewers nor I were able to access the code and data via the provided link, https://gitlab.univnantes.fr/petit-j-2/bnscore. Please ensure the link is working when resubmitting the revised version of your manuscript.

We look forward to receiving your revised manuscript.

Kind regards,

Patricia Wollstadt, Ph.D.

Academic Editor

PLOS ONE

Journal Requirements:

Reviewers' comments:

Reviewer's Responses to Questions

**Comments to the Author**

1. Is the manuscript technically sound, and do the data support the conclusions?

Reviewer #1: Yes

Reviewer #2: Yes

2. Has the statistical analysis been performed appropriately and rigorously? 

Reviewer #1: Yes

Reviewer #2: Yes

3. Have the authors made all data underlying the findings in their manuscript fully available?

Reviewer #1: No

Reviewer #2: No

4. Is the manuscript presented in an intelligible fashion and written in standard English?

Reviewer #1: Yes

Reviewer #2: Yes

5. Review Comments to the Author

Reviewer #1: **Summary

The paper investigates the perceived collision risk of a self-driving car’s passenger when driving in shared traffic spaces and encountering pedestrians crossing the way. Concretely, the authors evaluate the perceived risk by varying seven factors of the dynamics of the driving situation in a driving simulator. The user’s perceived risk is then measured by an electrodermal activity sensor and an analogue potentiometer allowing the user to indicate its feeling of risk. With Bayesian networks, the authors finally are able to analyze the relationship of the driving situation factors with the measured perceived risk values.

**Strengths

The paper is very well written, and the conclusions of the analysis are supported well by the data. We would like to thank the authors for their submission. The paper evaluates in detail pedestrian-vehicle interactions and helps the research community to understand the factors that influence the passenger’s feeling of an autonomous vehicle. We therefore recommend an acceptance with minor revisions.

**Weaknesses

The authors already published similar work that shows the relationship between some driving situation factors based on Bayesian networks. The contribution is only a wider evaluation with seven factors that are analyzed. The scientific value of the paper lies only on the experimental additions.

A discussion on how objective risk models, such as the mentioned TTI, relate to the findings with the EDA are given. However, it would still be interesting to compare the risk value from TTI quantitatively and show how the perceived risk could improve TTI measures in pedestrian-vehicle interactions in more detail.

Finally, we could not access the provided link with the data supporting the paper. Please provide the data for the paper.

**Minor revisions

1. The background and reasons for the choice of the used methods are not given in sufficient detail. It should be described or rewritten in a more understandable way why Bayesian networks were chosen for the evaluation of the relationship of the driving situation factors with the perceived risks, why the factors were treated as bi-modal random variables and why a Tweedie distribution was chosen to model the electrodermal activity responses indicators (see page 16).

2. The v-speed variation is only in between the range of 20-30 kph. This could have an effect that the factor v-speed is less influencing the perceived risk value. We recommend varying the velocity between 10 – 50 kph because this is realistic for shared space scenarios or to describe this circumstance in the experiments.

3. It should be discussed what the sim-to-real gap is for the experiments. For example, the modeling of the pedestrians walking, or facial features could be deviating from reality. These can change the results of the experiments.

4. We recommend changing the title to “Perceived Risk Assessment and Associated Electrodermal Activity of a Self-Driving Car Passenger in an Urban Shared Space” to highlight the perceived risk assessment because there are many works for objective risk assessment, such as TTI, which is not analyzed in this paper.

**Other comments

1. Please rewrite the note description in Figure 3 on page 12, it is currently hard to understand.

2. Figure 5 is referred to as Figure 5A on page 14. There is no sub-figure A.

3. The abbreviation BIC score is not written out when used the first time.

Reviewer #2: ## Content:

Dependence between risk metrics:

You mention early that there are indicators for dependence between risk indicators in prior work (Petit et al. 2021). However, in the subsequent argument you mainly appear to be concerned with their complementarity, e.g., different sensitivities to low-risk situations (I assume as an argument for using both), than with possible downsides of the dependence for your investigation. When reading that part I wondered whether it might not be a methodological issue to let the active thinking about risk for the real-time subjective evaluation bias overall risk perception (and skin conductance).

On a similar note, I believe that the four crash situations may have strongly altered alertness. This might have eliminated over-trust, which may be observable within the population or specific user groups. In the discussion a more critical examination of such introduced biases would be appreciated.

I do appreciate your style of data analysis and that this analysis does indeed collect some of the questions that I had when reading about the study setup.

4.1

It would be great if you could cite some examples on dynamic human–machine interfaces to improve collision risk estimation that could be tested using your measures.

## Language & style:

Introduction: Related work all presented in present tense. Past tense would often be more appropriate and has been correctly used in, e.g., Section 1.3.

Figure 3: Use larger axis labels.

Section 1.2: "Studies have proven"

Studies cannot prove but rather indicate, suggest, etc.. A proof belongs to logic, mathematics, and perhaps law.

2.3 "intercepted" -> "intersected"

Some explanations are given multiple times. Such redundancy can be reduced easily (e.g., Type 1 and type 2 process explanations).

## PLOS requirements:

3. Download link for data given in paper but did not appear to be functional when tested (no connection to server)

6. PLOS authors have the option to publish the peer review history of their article (what does this mean?). If published, this will include your full peer review and any attached files.

Reviewer #1: No

Reviewer #2: No

---

## [Author Response · Author response to Decision Letter 0]

15 Jun 2023

You will find below a point-by-point response in the corresponding file.

---

## [Editor Report · Decision Letter 1]

31 Jul 2023

Subjective Risk and Associated Electrodermal Activity of a Self-Driving Car Passenger in an Urban Shared Space

PONE-D-23-10461R1

Dear Dr. Mars,

We’re pleased to inform you that your manuscript has been judged scientifically suitable for publication and will be formally accepted for publication once it meets all outstanding technical requirements.

Kind regards,

Patricia Wollstadt, Ph.D.

Academic Editor

PLOS ONE
---

## [Editor Report · Acceptance letter]

12 Sep 2023

PONE-D-23-10461R1 

Subjective Risk and Associated Electrodermal Activity of a Self-Driving Car Passenger
in an Urban Shared Space 

Dear Dr. Mars:

I'm pleased to inform you that your manuscript has been deemed suitable for publication in PLOS ONE. Congratulations! Your manuscript is now with our production department. 

Kind regards, 

on behalf of

Dr. Patricia Wollstadt 

Academic Editor

PLOS ONE